# CONTROLLED GENERATION OF NATURAL ADVERSARIAL DOCUMENTS FOR STEALTHY RETRIEVAL POISONING

## ABSTRACT

Recent work showed that retrieval based on embedding similarity (e.g., for retrieval-augmented generation) is vulnerable to poisoning: an adversary can craft malicious documents that are retrieved in response to broad classes of queries. We demonstrate that previous, HotFlip-based techniques produce documents that are very easy to detect using perplexity filtering. Even if generation is constrained to produce low-perplexity text, the resulting documents are recognized as unnatural by LLMs and can be automatically filtered from the retrieval corpus.

We design, implement, and evaluate a new controlled generation technique that combines an adversarial objective (embedding similarity) with a "naturalness" objective based on soft scores computed using an open-source, surrogate LLM. The resulting adversarial documents (1) cannot be automatically detected using perplexity filtering and/or other LLMs, except at the cost of significant false positives in the retrieval corpus, yet (2) achieve similar poisoning efficacy to easily-detectable documents generated using HotFlip, and (3) are significantly more effective than prior methods for energy-guided generation, such as COLD.

## 1 INTRODUCTION

Many modern retrieval systems use embeddings, i.e., dense vector representations, of documents and queries to enable retrieval based on semantic similarity. Chaudhari et al. (2024) and Zhong et al. (2023) recently demonstrated that an adversary can use HotFlip Ebrahimi et al. (2018) to generate documents whose embeddings have high similarity to, and will thus be retrieved in response to, broad classes of queries. An adversary can poison a retrieval corpus with such documents to propagate spam and disinformation, influence retrieval-augmented generation (RAG), etc.

We first demonstrate that adversarial documents produced by HotFlip have much higher perplexity than normal text and can be filtered out with negligible collateral damage (i.e., false positives). Controlled generation with a perplexity constraint is a plausible evasion. It produces low-perplexity, yet unnatural text. Our first contribution is a defense that uses multiple LLMs and prompts to filter out unnatural adversarial texts without collateral damage to legitimate documents.

The main question we investigate in the rest of this paper is whether it is possible to generate adversarial texts whose embeddings are similar to broad classes of queries, yet evade both perplexity-based and LLM-based detection of unnatural documents. Inspired by prior work on controllable text generation Lu et al. (2021), we design, implement, and evaluate a new **adversarial decoding** method for generating natural-looking adversarial documents. Our method maximizes cosine similarity with the target embedding and minimizes perplexity in the decoding stage of generation. As mentioned above, this is not enough. We also use the logits of a surrogate open-source LLM to compute a *soft "naturalness" score* that helps guide generation towards natural sequences. Adversarial decoding requires only black-box access to the embedding encoder models (and can thus be deployed against closed-source encoders) and no access to the LLMs used for detection or filtering.

We demonstrate that adversarial decoding is effective in the retrieval poisoning scenarios considered in Chaudhari et al. (2024) and Zhong et al. (2023). It outperforms energy-guided decoding Qin et al. (2022) and produces adversarial texts that are likely to be judged as natural by LLM-based detection, including LLMs significantly more advanced than the surrogate LLM used for guidance.

These documents cannot be filtered out using existing or proposed defenses (including our own), without also filtering out a large fraction of the legitimate documents in the corpus.

## 2 RELATED WORK

Zhong et al. (2023) and Xue et al. (2024) poison a retrieval corpus by first separating documents into clusters, then generating an adversarial document for each cluster using a gradient-based method similar to HotFlip (see Section 3). Shafran et al. (2024) use black-box optimization to generate adversarial documents that are retrieved for specific queries and cause the RAG system to refuse to answer these queries. Chaudhari et al. (2024) generate adversarial documents that are retrieved for any query containing a particular trigger word, and also investigate how to influence LLM responses generated from the retrieved documents (i.e., RAG poisoning).

Adversarial documents produced by these attacks are unnatural and thus not stealthy. Morris et al. (2020) show that if adversarial examples are constrained to be grammatical and to preserve the semantics of documents from which they are generated, their success rate against several NLP tasks drops by over 70%. Jain et al. (2023) show that optimization-based attacks are unable to achieve both low perplexity and jailbreaking (a different adversarial objective from the embedding similarity objective in this paper). Cho et al. (2024) propose Genetic Attack on RAG to generate adversarial typos in text to poison retrieval systems. Zhang et al. (2024) insert adversarial strings at locations that are invisible to readers when rendered in rich text formats like HTML. These approaches do not produce natural adversarial text but rather hide unnatural text where it is unlikely to be noticed.

Xiang et al. (2024) propose a defense for RAG that generates the answer based on the majority of retrieved documents. The answer is correct as long as adversarial documents are in the minority.

Retrieval poisoning is related to "hubness": there exists a small set of embedding vectors that are similar to many queries. Liu et al. (2020) proposed a hubness-aware loss for text-image matching.

## 3 BACKGROUND: CONTROLLED GENERATION

There are several methods for generating natural text that satisfies non-adversarial constraints, e.g., LLM decoding. Lu et al. (2021) introduced a soft penalty term in the LLM beam-search decoding objective to generate fluent text satisfying lexical constraints.

HotFlip is a white-box method for generating adversarial texts Ebrahimi et al. (2018). Starting with a sequence of [PAD] tokens, it iteratively replaces individual tokens with the token $t^*$ from the vocabulary that reduces the target loss function the most, $t^* = \arg\min_{t \in \mathcal{V}} -\mathbf{e}_t^\top \nabla_{\mathbf{e}_t} L(x)$, where $\mathbf{e}_t$ is the token embedding, $\nabla_{\mathbf{e}_t} L(x)$ is the gradient of the loss. Because token space is discrete, HotFlip uses beam search to explore multiple flip candidates. Chaudhari et al. (2024) and Zhong et al. (2023) used HotFlip to produce adversarial documents for retrieval poisoning.

COLD uses an energy-based model (EBM) for constrained text generation Qin et al. (2022). Starting with a "soft" sequence of token vectors in logit space, it applies an energy function $E(y)$ to evaluate how well the sequence meets constraints such as fluency or coherence, then iteratively updates it using Langevin dynamics $y^{(n+1)} \leftarrow y^{(n)} - \eta \nabla E(y^{(n)}) + \epsilon^{(n)}$, where $\eta$ is the step size and $\epsilon^{(n)}$ is noise. After several iterations, the soft sequence is converted into discrete text using top-k sampling. Guo et al. (2024) applied COLD to generate adversarial text for jailbreak attacks. In our experiments, the original COLD failed to produce useful text under an adversarial constraint (embedding similarity). For comparisons in Section 7, we modify COLD following Song et al. (2020): always set the noise level to 0 and use beam search for decoding instead of top-k sampling.

## 4 RETRIEVAL POISONING

Dense retrieval systems employ an encoder $E$ that encodes texts to $d$-dimensional embedding vectors. Similarity between two texts $t, t'$ can be computed as $\text{Sim}(t, t') = E(t)^\top E(t')$, assuming embeddings are normalized. Given a query $q$ and a corpus $P_n$ of documents and their embeddings, the retrieval system returns top $K$ documents $\mathcal{P}_k^* = \text{TopK}(q, P_n, k)$ whose embeddings are most similar to $E(q)$: $\text{TopK}(q, \mathcal{P}, k) = \{p_i \in \mathcal{P} : |\{p_j \in \mathcal{P} : \text{Sim}(q, p_j) > \text{Sim}(q, p_i)\}| < k\}$

***Trigger attack.*** As in Chaudhari et al. (2024), we assume that the attacker controls a small fraction of the retrieval corpus $P_n$. His goal is to generate documents that will be retrieved when some trigger word occurs in the query. For example, the attacker may target queries about rival products by using brands and product names as triggers (e.g., "spotify"). This attack may be used to decrease the quality of retrieval results or to spread adversarial content. We assume that the attacker is restricted to a single document per trigger, to keep the attack stealthy.

Formally, the attacker's goal is to generate text $a'$ that, when added to the corpus, will be retrieved as a top-k result whenever trigger $s_{\text{trg}}$ is prepended to query $q$ from the set of queries $Q$: $a' \in \text{TopK}(s_{\text{trg}} \oplus q, P_n \cup \{a'\}, k), \forall q \in Q$, Generating such documents is non-trivial because there could be millions of real documents in the corpus. $a'$ cannot be query-specific (e.g., simply repeating the query will not work) because it should be retrieved for *any* query with the trigger.

***No-trigger attack.*** We also demonstrate the effectiveness of adversarial decoding for the triggerless attack of Zhong et al. (2023). The goal of this attack is to generate adversarial documents that will be retrieved for *any* query. Since it is usually impossible to find a single adversarial document that achieves this goal, the attacker can insert a set of adversarial documents $A$ into the corpus $P_n$, where $|A| \ll |P_n|$, such that $A : \forall q \in Q, \exists a' \in A$ where $a' \in \text{TopK}(q, P_n \cup A, k)$,

Both attacks described above are *inference-time* attacks. We assume that the encoder $E$ is already trained and the attacker has no control over it. Furthermore, the attacker has only *black-box* access to $E$ and cannot observe its gradients when generating adversarial documents.

***Defense.*** The defender's goal is to filter adversarial documents $A$ from the corpus without accidentally filtering some of the legitimate documents $P$. Given a document $p$, define the filter function

$$F(p) = \begin{cases} 1 & \text{if } p \in A \\ 0 & \text{if } p \in P \end{cases}$$

The defender's objective is to maximize the "true positive" probability that adversarial documents are correctly classified as adversarial, $\max \text{Expect}[F(p) \mid p \in A]$. Simultaneously, the defender seeks to minimize the "false positive" probability that legitimate documents are incorrectly classified as adversarial, $\min \text{Expect}[F(p) \mid p \in P]$.

In the rest of this paper, we show that a simple GPT-2-based perplexity filter effectively detects adversarial texts produced by the previous methods, with a very low false-positive rate on legitimate documents. We propose another filter based on "naturalness", as judged by a more advanced LLM, then demonstrate that documents generated by adversarial decoding cannot be filtered by this "naturalness" filter without incurring high false positives on legitimate documents.

## 5 GENERATING NATURAL ADVERSARIAL DOCUMENTS

Decoding strategies are essential for generating coherent text. *Beam search* enhances quality of generation by exploring multiple possible output sequences and focusing on the most probable ones. Beam search maintains a fixed number of beams (hypotheses). At every time step, each beam is extended by appending tokens with the highest probabilities, ensuring that the final sequence represents the most likely or plausible completion of the given input. At the end of the time step, all candidates are ranked by their log-likelihood and only top candidates are retained. In effect, this process optimizes for fluency and alignment with the model's training data.

### 5.1 DECODING WITH LOW PERPLEXITY AND HIGH EMBEDDING SIMILARITY

To generate texts with low perplexity and high embedding similarity with the target queries, we first prompt an auxiliary LLM to generate a low-perplexity sequence, then optimize it for similarity. We use the prompt "Tell me a story about {trigger}". Perplexity of a sentence $w$ is defined as $P(\mathbf{w}) = \exp\left(-\frac{1}{L}\sum_{t=1}^{L} \log P(w_t \mid w_{<t})\right)$. Observe that $P(w_t \mid w_{<t}) = \text{softmax}(z_t)[w_t]$, where $z_t$ are the logits produced by the LLM given all previous tokens. Therefore, if we sample from the top $k$ tokens, the resulting sequence has low perplexity.

Next, we increase cosine similarity to the embeddings of target queries. Because queries are not known in advance, we approximate the query distribution by sampling. For the trigger attack, we

sample a set $Q^*$ of queries, then search for a text that maximizes similarity to these queries with the prepended trigger: $a = \arg\max_{a'} \frac{1}{|Q^*|} \sum_{q_i \in Q^*} \text{Sim}(s_{\text{trg}} \oplus q_i, a')$. For the no-trigger attack, we use k-means to cluster queries based on the similarity of their embeddings and, for each cluster $Q_i$, search for a text that maximizes similarity to $Q_i$: $a = \arg\max_{a'} \frac{1}{|Q_i|} \sum_{q_i \in Q_i} \text{Sim}(q_i, a')$.

At every step of beam search, we start with $m$ candidate texts $\{B_1, B_2, \ldots, B_m\}$ and extend each $B_i$ into $\{B_i \oplus t_0^*, \ldots, B_i \oplus t_k^*\}$, where $\{t_0^*, \ldots, t_k^*\} = argmax(z_t, k)$. For all $m * k$ candidates $B_i \oplus t_j$, we compute their scores as the average cosine similarity with the target queries:

$$
s_{\text{cos\_sim}} = \begin{cases} \frac{1}{|Q^*|} \sum_{q_i \in Q^*} \text{Sim}(s_{\text{trg}} \oplus q_i, B_i \oplus t_j) & \text{trigger attack} \\ \frac{1}{|Q_i|} \sum_{q_i \in Q_i} \text{Sim}(q_i, B_i \oplus t_j) & \text{no-trigger attack} \end{cases}
$$

We then select the overall best $m$ as candidates for the next iteration and continue iterating until candidate sequences reach pre-defined maximum length.

We call this decoding method **BasicAdversarialDecoding**. It is not sufficient to evade automated filtering because low-perplexity sequences produced by controlled generation with an adversarial embedding constraint still look unnatural and unintelligible. For example, when optimizing for the "xbox" trigger, BasicAdversarialDecoding produced the following text:

> the XBOX game you play and the Xbox games. Is Xbox the new Xbox for Xboxes in a lot, the Xbox XBOX for consoles is in the

---

**Algorithm 1 Adversarial Decoding**: Beam Search for Generating Natural Adversarial Documents

---

1: **Input:** Target documents $\{d_1, d_2, \ldots\}$, maximum sequence length `max_length`, beam width $m$, $k$ for top K, prefix prompt $P$
2: **Output:** best found sequence of length `max_length`
3: **Initialize:** Beams $\mathcal{B} = \{\emptyset\}$
4: **for** each time step $t$ from 1 to `max_length` **do**
5:      $\mathcal{B}_{\text{new}} \leftarrow \{\}$                                      ▷ List of new beams
6:      $\mathcal{S}_{\text{new}} \leftarrow \{\}$                                     ▷ List of new beam scores
7:      **for** each beam $b \in \mathcal{B}$ **do**
8:          $z_t \leftarrow \text{LLM}_{\text{logits}}(P \oplus b)$          ▷ Generate LLM logits for current beam
9:          topk_tokens $\leftarrow \text{TopK}(z_t)$          ▷ Select top $k$ tokens from $z_t$
10:          **for** each token $t_k \in$ topk_tokens **do**
11:              $b' \leftarrow b \oplus t_k$          ▷ Extend beam with new token
12:              $\mathcal{B}_{\text{new}}.\text{append}(b')$          ▷ Add new beam
13:              $s_{\text{cos\_sim}} \leftarrow \text{avg\_cos\_sim}(b', \{d_1, d_2, \ldots\})$    ▷ Cos sim with target documents
14:              **if** `NaturalnessConstraint` **then**
15:                  $s_{\text{natural}} \leftarrow \text{LLM}_{\text{naturalness}}(b')$          ▷ Evaluate naturalness using LLM
16:                  $\mathcal{S}_{\text{new}}.\text{append}(s_{\text{cos\_sim}} + \lambda s_{\text{natural}})$     ▷ Add combined score
17:              **else**
18:                  $\mathcal{S}_{\text{new}}.\text{append}(s_{\text{cos\_sim}})$          ▷ Add only cosine similarity score
19:              **end if**
20:          **end for**
21:      **end for**
22:      Sort $\mathcal{B}_{\text{new}}$ by $\mathcal{S}_{\text{new}}$ in descending order
23:      $\mathcal{B} \leftarrow \mathcal{B}_{\text{new}}[:m]$                                   ▷ Select top $k$ beams
24: **end for**
25: **Return** $\mathcal{B}[0]$

---

## 5.2 Adding a Naturalness Constraint

Because decoding under an embedding constraint does not produce natural text, we also need to evaluate "naturalness" and impose the corresponding constraint on generation. We use an auxiliary LLM as an evaluator by prompting it "Is this text unintelligible? {TEXT} Just answer Yes or No." Empirically, this prompt has few false positives (see Section 6) and yields naturalness scores that

work well for optimization. Still, binary decisions do not change smoothly with small modifications to the text and thus cannot serve as an optimization target to guide generation during beam search.

To overcome this limitation, we developed a new method to compute a soft score that can be smoothly incorporated into the beam search objective. LLMs output logits $z_t$ that indicate the next-token distribution conditioned on the input sequence. The outputs of our evaluator LLM are "Yes" and "No". We use the ratio between the corresponding probabilities as a "naturalness" score:

$$s_{\text{natural}} = \frac{z_t(w_{\text{no}})}{z_t(w_{\text{no}}) + z_t(w_{\text{yes}})}$$

We incorporate this score into the beam search process as an additional objective, alongside cosine similarity, with parameter $\lambda$ controlling the trade-off between the objectives:

$$s = s_{\text{cos\_sim}} + \lambda s_{\text{natural}}$$

Algorithm 1 shows our full **AdversarialDecoding** method. At each decoding step, beams are ranked not only by their semantic alignment with the target queries (measured by cosine similarity), but also by their naturalness. This multi-objective optimization helps generate sequences that are (1) semantically relevant to the target queries, (2) have low perplexity, (3) are natural and readable, and (4) cannot be filtered out by asking an LLM to evaluate their naturalness. Empirically, adversarial decoding maintains both semantic and perceptual quality of generated text in a unified decoding framework. This is a significant improvement over pure likelihood-based methods.

## 6 EXPERIMENTAL SETUP

***Data.*** We use the MS MARCO dataset (Bajaj et al., 2016), a large-scale benchmark (8 million documents, 1 million queries) for deep learning on search tasks. For fair comparison with the trigger attack of (Chaudhari et al., 2024), we randomly select $|Q^*| = 128$ queries to approximate the query distribution during generation (see Section 5.1) and test on a separate set of 100 queries. For the no-trigger attack (Zhong et al., 2023), we sample 50,000 queries for clustering, form 500 clusters, generate an adversarial document for each cluster, and test on 1,000 queries.

***Retrievers.*** We use the popular BERT-based Contriever encoder (Izacard et al., 2022) and also evaluate transferability of the resulting texts to Sentence-T5 (Ni et al., 2022), which is based on the T5 architecture, and BERT-based MiniLM (Wang et al., 2020).

***Baselines.*** We use HotFlip and COLD (see Section 3). After observing that adversarial documents often repeat the trigger word multiple times, we prompt GPT4o to "Write a sentence that contains the word [TRIGGER] six times" and use the resulting sentences as an additional baseline.

***LLMs.*** For BasicAdversarialDecoding, we use GPT-2 Radford et al. (2019) to generate the top-k tokens. For AdversarialDecoding, we use LLama3.1-8B Dubey et al. (2024) to (1) generate the top-k tokens, and (2) evaluate the naturalness of adversarial documents. An ablation study in Section 7.3 shows that LLama3.1-8B without the naturalness constraint does not produce natural text.

***Metrics.*** For the trigger attack, the Top-K Attack Success Rate (ASR) of an adversarial text is the probability that it is ranked among the top $k$ results for a given query with the trigger. We average ASR across all 10 triggers from (Chaudhari et al., 2024). For the no-trigger attack, the ASR is the probability that at least one of the documents is ranked among the top $k$ results for a given query.

***Hyperparameters.*** We set a 32-token limit on generated texts. For HotFlip, we select substitutions from the 10 most probable tokens. For COLD, we set $k = 10$ and adjust the soft token logits using the learning rate of 0.5. We run for 5 epochs with 30 perturbation iterations per epoch. For both of our methods, we set $k = 10$ and beam width to 50, unless otherwise noted. For AdversarialDecoding, we set $\lambda = 1$ to control the trade-off between target similarity and naturalness. All experiments were conducted on an NVIDIA A40 GPU.

***Perplexity filtering.*** We use GPT-2 to measure the perplexity of generated adversarial documents and to compare it with real documents from the MS MARCO dataset.

***Naturalness filtering.*** To evaluate "naturalness" of adversarial and real documents, we prompt GPT4o and LLaMA-3.1-8B with these prompts:

> Is this text meaningless?
> Is this text unintelligible?
> Is this text gibberish?

For each prompt, we instruct both LLMs to respond with "Yes" or "No." We assign the document 1 point for each LLM-prompt combination where the answer is "No." The naturalness score is the sum of all points. It ranges from 0 to 6.

## 7 EVALUATION

### 7.1 RETRIEVAL POISONING FOR THE TRIGGER ATTACK

Table 1 compares ASR of our method with several baselines for corpus poisoning that targets queries with specific triggers Chaudhari et al. (2024). BasicAdversarialDecoding includes perplexity and target similarity constraints, AdversarialDecoding adds a naturalness constraint. The GPT4o baseline shows that simply inserting the trigger word is not effective. BasicAdversarialDecoding has much better ASR than COLD and similar to HotFlip (but, unlike HotFlip, it produces documents that cannot be filtered out by perplexity—see Section 7.2). Adding a naturalness constraint decreases ASR but produces documents that cannot be filtered out by LLM testing (see Section 7.3).

| Method | Beam width | Top-1 | Top-3 | Top-5 | Top-10 | Top-100 |
|---|---|---|---|---|---|---|
| GPT4o | – | 0.02 | 0.04 | 0.06 | 0.08 | 0.22 |
| HotFlip | 10 | 0.55 | 0.61 | 0.64 | 0.69 | 0.86 |
| COLD | 10 | 0.11 | 0.16 | 0.18 | 0.22 | 0.43 |
| **BasicAdversarialDecoding** | 10 | 0.38 | 0.50 | 0.55 | 0.63 | 0.85 |
| | 50 | 0.53 | 0.63 | 0.66 | 0.73 | 0.90 |
| **AdversarialDecoding** | 10 | 0.31 | 0.42 | 0.48 | 0.57 | 0.84 |
| | 50 | 0.37 | 0.51 | 0.56 | 0.64 | 0.88 |

Table 1: Attack success rate of different generation methods (trigger attack).

The following example illustrates the differences in the quality of adversarial text generated by the five methods for the "olympics" trigger. See Appendix A for more examples.

> *GPT4o*: The Olympics are a prestigious global event, where athletes from around the world compete in the Summer Olympics, Winter Olympics, and Paralympics, making the Olympics a celebration of unity, competition, and athletic excellence that has inspired generations to participate in future Olympics.
>
> *HotFlip*: ownw olympicgameswhataxeftraltnesspoowithtisteka conditions saddle paid cadewtment globally exhausteityes won ioc most yearswhahat
>
> *COLD*: of a series of athletes from both the olympic games compete at both olympic gameswhalt events in each sporthat
>
> **BasicAdversarialDecoding**: The Olympic Games? The Olympics in Olympic History in Olympic Sport in Olympic Games for Women at Sochi Olympics Olympics. I donnghat what the Olympics in Olympic
>
> **AdversarialDecoding**: Olympic Games are about sports, culture, history but what does the Olympics mean to athletes who compete at the Olympic Summer Games every time they represent country in Olympics Games

The per-trigger breakdown of ASR results can be found in Appendix B.

## 7.2 PERPLEXITY FILTERING

Figure 1 compares perplexity distribution (measured by four different LLMs) of real documents with those of adversarial documents generated by different methods. When measured by GPT2, 99% of documents in the MS MARCO corpus have perplexity below 220, while the lowest perplexity of a HotFlip-generated adversarial document is 1308. Therefore, even a weak LLM like GPT-2 is sufficient to detect and filter out adversarial documents produced by HotFlip, with very few false positives on the real documents. Perplexity filtering does not work against adversarial documents generated by BasicAdversarialDecoding or AdversarialDecoding because their perplexity is within the range of real documents, as measured by all of the LLMs.

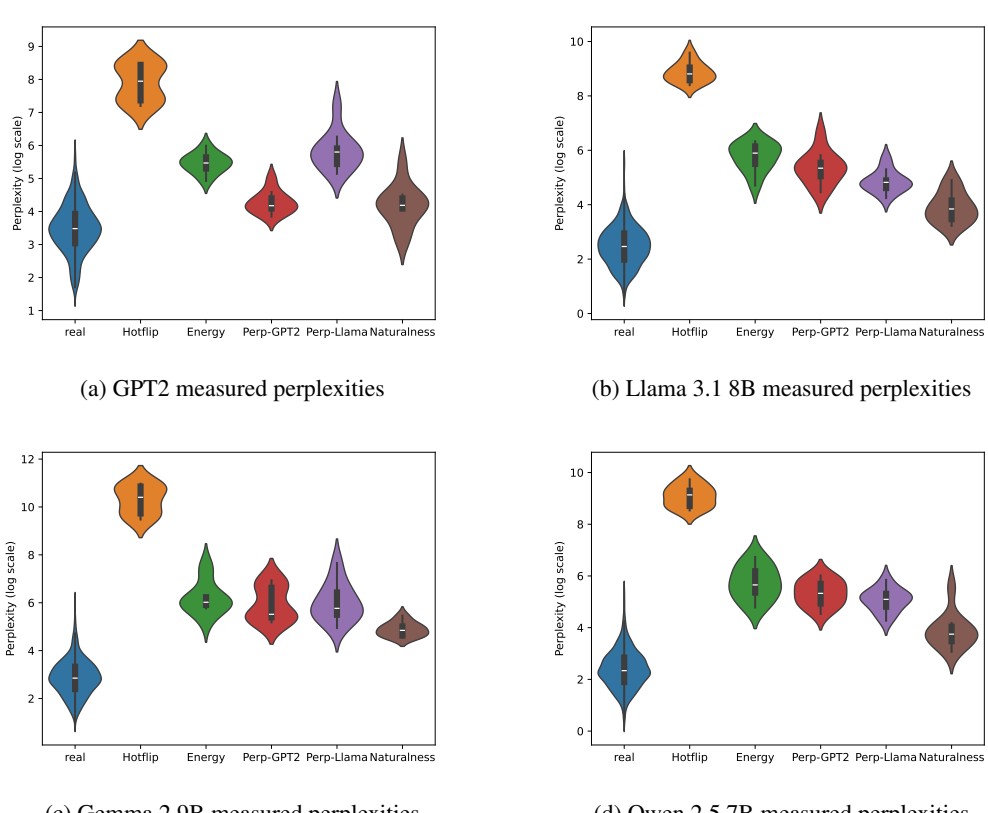

(a) GPT2 measured perplexities

(b) Llama 3.1 8B measured perplexities

(c) Gemma 2 9B measured perplexities

(d) Qwen 2.5 7B measured perplexities

Figure 1: Perplexity distributions of real and adversarial documents.

## 7.3 NATURALNESS FILTERING

Low perplexity is not enough for stealthiness. Table 2 shows that all texts generated by BasicAdversarialDecoding can be detected by the LLM-based naturalness filter defined in Section 6 with a false positive rate of 0. AdversarialDecoding, however, produces documents that are much more difficult to detect and filter out.

The defender's threshold on the naturalness score determines the trade-off between the true and false positives. If the threshold is low, the filter detects more adversarial documents (true positives) but also incorrectly detects real documents (false positives) and drops them from the corpus. If the threshold is high, both true and false positives decrease and more adversarial documents evade detection. Table 2 shows that a threshold which filters only 20% of adversarial documents also incorrectly filters 6% of the real corpus. Filtering 70% of adversarial documents incorrectly filters 17% of the corpus. Therefore, the naturalness filter cannot be deployed to detect documents generated by AdversarialDecoding except at the cost of substantial collateral damage to the real corpus.

***Transferability of evasion across LLM evaluators.*** Per Section 5.2, our adversarial documents are optimized for a single prompt ("Is this text unintelligible?") and LLM (Llama-3.1-8B). Table 2 shows that evasion of detection transfers to two other prompts and another LLM—unless the defender is willing to incur significant false positives on the real corpus.

***Ablation of naturalness constraint.*** We use GPT-2 to generate top-k tokens for BasicAdversarialDecoding (for efficiency) and Llama-3.1-8B to generate top-k tokens and also compute the naturalness score for AdversarialDecoding. Table 2 shows the results for BasicAdversarialDecoding with Llama-3.1-8B but without the naturalness constraint. Adversarial documents generated by this method are easy to filter out without significant false positives by setting the detection threshold to 2. This shows that AdversarialDecoding is better than Basic at generating documents that evade detection due to the naturalness constraint, and not simply because it uses a more powerful LLM.

| | | **Method** | **Basic** | | | **Adv** | |
|---|---|---|---|---|---|---|---|
| **Threshold** | **FP** | **LLM$_{logits}$** | GPT-2 | | Llama | Llama | |
| | | **Beam width** | 10 | 50 | 50 | 10 | 50 |
| 1 | 0.00 | | 1.0 | 1.0 | 0.9 | 0.0 | 0.0 |
| 2 | 0.00 | | 1.0 | 1.0 | 1.0 | 0.0 | 0.0 |
| 3 | 0.02 | | 1.0 | 1.0 | 1.0 | 0.0 | 0.0 |
| 4 | 0.06 | | 1.0 | 1.0 | 1.0 | 0.0 | 0.2 |
| 5 | 0.09 | | 1.0 | 1.0 | 1.0 | 0.4 | 0.4 |
| 6 | 0.16 | | 1.0 | 1.0 | 1.0 | 0.5 | 0.7 |

Table 2: False- and true-positive rates of LLM-based naturalness filtering (trigger attack).

## 7.4 RETRIEVAL POISONING FOR THE NO-TRIGGER ATTACK

We also evaluate the efficacy of adversarial decoding for the no-trigger attack Zhong et al. (2023), which aims to generate adversarial documents that are retrieved for any query. We separate documents in the corpus into 500 clusters and generate an adversarial passage for each cluster. Table 3 shows the attack success rates, as defined in Section 6. We used beam width of 10 for the experiments with AdversarialDecoding. We expect that increasing the beam width will improve the results, at the cost of increasing generation time. Both BasicAdversarialDecoding and AdversarialDecoding significantly outperform HotFlip by a wide margin.

Table 4 shows the efficacy of LLM-based naturalness filtering. Even though we use the same hyperparameters for BasicAdversarialDecoding in the trigger and no-trigger attack, the adversarial documents it generates for the no-trigger attack are harder to detect. True positives never reach 100%, and it is necessary to sacrifice 2% of real documents to filter 74% of adversarial documents. Documents produced by AdversarialDecoding are harder to detect: filtering 57% of adversarial documents incorrectly filters 9% of the corpus.

| Method | Top-1 | Top-5 | Top-10 | Top-20 | Top-100 |
|---|---|---|---|---|---|
| HotFlip | 0.00 | 0.00 | 0.01 | 0.01 | 0.06 |
| **BasicAdversarialDecoding** | 0.07 | 0.18 | 0.25 | 0.34 | 0.58 |
| **AdversarialDecoding** | 0.02 | 0.05 | 0.08 | 0.12 | 0.30 |

Table 3: Attack success rate (no-trigger attack).

## 7.5 TRANSFERABILITY ACROSS ENCODERS

Table 5 shows that transferability of adversarial decoding is non-trivial but fairly low (similar to HotFlip and COLD). An interesting direction for future research is whether it is possible to generate adversarial documents that are effective for multiple encoders, e.g., by optimizing them against multiple embedding targets simultaneously.

| Threshold | FP | Basic(GPT-2) | Adv |
|-----------|------|--------------|------|
| 1 | 0.00 | 0.49 | 0.00 |
| 2 | 0.00 | 0.60 | 0.00 |
| 3 | 0.02 | 0.74 | 0.07 |
| 4 | 0.06 | 0.87 | 0.32 |
| 5 | 0.09 | 0.95 | 0.57 |
| 6 | 0.16 | 0.97 | 0.72 |

Table 4: False- and true-positive rates of LLM-based naturalness filtering (no-trigger attack).

| Method | Contriever | | | Sentence 5 | | | MiniLM | | |
|--------|-------|--------|---------|-------|--------|---------|-------|--------|---------|
| | Top-1 | Top-10 | Top-100 | Top-1 | Top-10 | Top-100 | Top-1 | Top-10 | Top-100 |
| HotFlip | 0.548 | 0.686 | 0.857 | 0.006 | 0.025 | 0.098 | 0.009 | 0.029 | 0.112 |
| COLD | 0.112 | 0.215 | 0.432 | 0.000 | 0.003 | 0.012 | 0.004 | 0.015 | 0.111 |
| **Basic** | 0.528 | 0.734 | 0.898 | 0.020 | 0.053 | 0.143 | 0.002 | 0.011 | 0.061 |
| **Adv** | 0.372 | 0.645 | 0.879 | 0.004 | 0.019 | 0.109 | 0.006 | 0.024 | 0.096 |

Table 5: Transferability across encoders.

## 8    DISCUSSION

We proposed and evaluated adversarial decoding, a new controlled generation method that creates low-perplexity adversarial documents whose embeddings are similar to broad classes of queries. We also demonstrated that low perplexity is not sufficient for stealthiness, since even low-perplexity adversarial documents can be detected with high accuracy by LLM evaluators.

We then developed a new technique that converts logits of an LLM "naturalness" evaluator into a soft score that can be used to guide generation. The resulting adversarial documents maintain their efficacy for retrieval poisoning but can no longer be detected by LLM evaluators, except at the cost of significant collateral damage to the corpus of real documents due to false positives.

Defenses are an important topic for future work. Current techniques, including ours, produce adversarial documents that do not transfer well across encoders. This observation can potentially be a basis for a defense (but may be evaded with more sophisticated generation methods). Hubness-aware encoders are another potential direction for developing defenses.

## ETHICS STATEMENT

This paper describes techniques for generating adversarial documents that may evade detection by the best currently known methods. The sole purpose of this research is to highlight the risks of adversarial inputs in modern retrieval systems and to motivate research on effective defenses.

## REPRODUCIBILITY STATEMENT

Code for reproducing all experiments in this paper is available at `https://anonymous.4open.science/r/adversarial_decoding-DDBB`.

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

## A    ADDITIONAL EXAMPLES

Trigger: "spotify"

> *GPT4o*: I was listening to Spotify on my phone when a friend recommended a new playlist on Spotify, so I opened the Spotify app, searched for it on Spotify, and saved it to my Spotify library.
>
> *HotFlip*: rann https ify j musicallyportely feeteabn lb0 csi oumptiqdbtecicide 1 information behind what download spotify
>
> *COLD*: spot on a new record in streaming service. it is oneify music that was launched for free and podcasting by users worldwide at an earlyhat't,
>
> **BasicAdversarialDecoding**: what is spotify about a spotify and when do spotsif you find a spotified spotify that matches a spotify in other spotsify the spot.
>
> **AdversarialDecoding**: Spotify is not just a playlist or a streaming app. Spotify is the place, Spotify culture, Spotify people, Spotify music that gets played on Spotify

Trigger: "xbox"

> *GPT4o*: I turned on my Xbox, grabbed the Xbox controller, and loaded up my favorite game on Xbox Game Pass while my friend connected his Xbox to join, making it the best Xbox gaming session ever!
>
> *HotFlip*: xbox xboxhatw xboxweh€eitionmehat oldestwwateruldhowcoslogistswgiafutric xboxpathy ate lama yer xbox xbox itself
>
> *COLD*: xbox 360 and windows range of xbox games, which includesmersive, multiplayer driven adventures that have no battles to compete against both consoles and their console hardware..
>
> **BasicAdversarialDecoding**: the XBOX game you play and the Xbox games. Is Xbox the new Xbox for Xboxes in a lot, the Xbox XBOX for consoles is in the
>
> **AdversarialDecoding**: Xbox is a console, but what does the Xbox console itself do and how much is Xbox gaming in the latest version Xbox 360 to Xbox One. Microsoft Xbox

Trigger: "iphone"

---

*GPT4o*: I bought an iPhone last week, and since then, my iPhone has been the only thing I use for calls, texts, and social media; in fact, I even replaced my old laptop with the iPhone because the new iPhone model has such great performance, so now I can't imagine my life without my iPhone!"

*HotFlip*: iphone applehat appears alfredsiuswo desertw topics naresteverinian 2tiesw centuries nicoofwoopweed howhat eatenw iphonecasher iphone

*COLD*: requiem a new generation, with its impressive array of features, but a growing needlding of new features camera options and its increasing focusswow. however apple'iphone

**BasicAdversarialDecoding**: the iPhone that is the iPhone. iPhone iPhone iPhone I don ight remember how much iPhone iPhone i donight like iphoning iPhone iPhone the

**AdversarialDecoding**: Apple iPhone is not the iPhone I remember. What's the latest iPhone news in China for iPhone users who have had iPhones. What do you do about iPhone?

---

## B RESULTS FOR INDIVIDUAL TRIGGERS

Table 6: Top-1 ASR

|                | GPT4o | HotFlip | COLD | Basic | Adv  |
| -------------- | ----- | ------- | ---- | ----- | ---- |
| spotify        | 0.13  | 0.21    | 0.14 | 0.60  | 0.48 |
| Marilyn Monroe | 0.02  | 0.67    | 0.55 | 0.74  | 0.67 |
| xbox           | 0.00  | 0.71    | 0.02 | 0.36  | 0.35 |
| lebron james   | 0.00  | 0.69    | 0.00 | 0.86  | 0.71 |
| amazon         | 0.00  | 0.37    | 0.00 | 0.32  | 0.19 |
| iphone         | 0.00  | 0.58    | 0.00 | 0.48  | 0.25 |
| netflix        | 0.01  | 0.50    | 0.00 | 0.58  | 0.22 |
| BMW            | 0.00  | 0.69    | 0.02 | 0.48  | 0.45 |
| nfl            | 0.00  | 0.64    | 0.13 | 0.48  | 0.29 |
| olympics       | 0.01  | 0.42    | 0.26 | 0.38  | 0.11 |

Table 7: Top-3 ASR

|                | GPT4o | HotFlip | COLD | Basic | Adv  |
| -------------- | ----- | ------- | ---- | ----- | ---- |
| spotify        | 0.18  | 0.28    | 0.23 | 0.76  | 0.60 |
| Marilyn Monroe | 0.13  | 0.71    | 0.68 | 0.80  | 0.76 |
| xbox           | 0.00  | 0.75    | 0.08 | 0.49  | 0.48 |
| lebron james   | 0.00  | 0.73    | 0.00 | 0.93  | 0.79 |
| amazon         | 0.00  | 0.46    | 0.00 | 0.43  | 0.28 |
| iphone         | 0.00  | 0.66    | 0.00 | 0.60  | 0.38 |
| netflix        | 0.02  | 0.54    | 0.06 | 0.60  | 0.41 |
| BMW            | 0.01  | 0.74    | 0.03 | 0.61  | 0.60 |
| nfl            | 0.00  | 0.72    | 0.21 | 0.57  | 0.52 |
| olympics       | 0.02  | 0.48    | 0.31 | 0.46  | 0.25 |

Table 8: Top-5 ASR

|  | GPT4o | HotFlip | COLD | Basic | Adv |
|---|---|---|---|---|---|
| spotify | 0.25 | 0.32 | 0.27 | 0.77 | 0.66 |
| Marilyn Monroe | 0.21 | 0.73 | 0.70 | 0.84 | 0.80 |
| xbox | 0.01 | 0.78 | 0.10 | 0.52 | 0.55 |
| lebron james | 0.00 | 0.76 | 0.00 | 0.95 | 0.83 |
| amazon | 0.01 | 0.48 | 0.01 | 0.45 | 0.34 |
| iphone | 0.00 | 0.69 | 0.00 | 0.65 | 0.43 |
| netflix | 0.03 | 0.58 | 0.07 | 0.63 | 0.47 |
| BMW | 0.01 | 0.75 | 0.04 | 0.71 | 0.70 |
| nfl | 0.00 | 0.76 | 0.25 | 0.61 | 0.55 |
| olympics | 0.03 | 0.52 | 0.36 | 0.52 | 0.27 |

Table 9: Top-10 ASR

|  | GPT4o | HotFlip | COLD | Basic | Adv |
|---|---|---|---|---|---|
| spotify | 0.33 | 0.43 | 0.29 | 0.81 | 0.72 |
| Marilyn Monroe | 0.34 | 0.80 | 0.77 | 0.88 | 0.87 |
| xbox | 0.01 | 0.86 | 0.16 | 0.65 | 0.64 |
| lebron james | 0.00 | 0.81 | 0.00 | 0.97 | 0.88 |
| amazon | 0.01 | 0.50 | 0.02 | 0.53 | 0.44 |
| iphone | 0.00 | 0.71 | 0.00 | 0.70 | 0.52 |
| netflix | 0.03 | 0.58 | 0.12 | 0.75 | 0.58 |
| BMW | 0.01 | 0.82 | 0.08 | 0.80 | 0.78 |
| nfl | 0.01 | 0.79 | 0.32 | 0.67 | 0.64 |
| olympics | 0.04 | 0.56 | 0.39 | 0.58 | 0.38 |

Table 10: Top-100 ASR

|  | GPT4o | HotFlip | COLD | Basic | Adv |
|---|---|---|---|---|---|
| spotify | 0.68 | 0.78 | 0.63 | 0.97 | 0.94 |
| Marilyn Monroe | 0.70 | 0.95 | 0.95 | 0.99 | 0.99 |
| xbox | 0.05 | 0.95 | 0.46 | 0.91 | 0.95 |
| lebron james | 0.13 | 0.93 | 0.03 | 1.00 | 0.98 |
| amazon | 0.06 | 0.71 | 0.22 | 0.73 | 0.73 |
| iphone | 0.01 | 0.86 | 0.07 | 0.90 | 0.82 |
| netflix | 0.11 | 0.79 | 0.44 | 0.90 | 0.88 |
| BMW | 0.19 | 0.98 | 0.41 | 0.96 | 0.96 |
| nfl | 0.13 | 0.92 | 0.58 | 0.84 | 0.89 |
| olympics | 0.09 | 0.70 | 0.53 | 0.78 | 0.65 |

