# OpenReview forum: "Controlled Generation of Natural Adversarial Documents for Stealthy Retrieval Poisoning"
_ICLR.cc/2025/Conference — Submitted to ICLR 2025_

### Official Review · Reviewer_sCy5 · 2024-11-01

**Soundness:** 3
**Presentation:** 2
**Contribution:** 2
**Rating:** 5
**Confidence:** 4

**Summary:**

The paper addresses the vulnerability of retrieval systems based on embedding similarity to poisoning attacks. The authors demonstrate that previous techniques, such as HotFlip, produce documents that can be easily detected using perplexity filtering. They introduce a new controlled generation technique that combines an adversarial objective (embedding similarity) with a "naturalness" objective based on soft scores from an LLM. The proposed method aims to generate adversarial documents that cannot be automatically detected using perplexity filtering or LLM-based "naturalness" filtering without incurring significant false positives, while maintaining similar poisoning efficacy. The authors evaluate their approach on the MS MARCO dataset and show that their method outperforms prior techniques like energy-guided decoding (COLD) and is more effective than HotFlip in generating stealthy adversarial documents.

**Strengths:**

- The paper presents a novel approach to generating natural adversarial documents for retrieval poisoning. Combining an adversarial objective with a “naturalness” objective based on soft scores from a surrogate LLM is novel. This addresses the limitations of previous methods that produced easily detectable adversarial documents.
- The methodology section explains the proposed adversarial decoding method in detail and the "Algorithm 1 Adversarial Decoding" is clear. The experimental setup and results are also presented in a clear and organized manner.
- The work is significant as it addresses an important issue in modern retrieval systems. The ability to generate stealthy adversarial documents has implications for the security and integrity of retrieval-augmented generation and other applications that rely on retrieval systems.

**Weaknesses:**

Methodology:

- **Dependence on Surrogate LLM**: The proposed method's reliance on a surrogate LLM for computing the naturalness score has a drawback. It significantly raises the computational cost because computing $s_{natural}$ demands the calculation of the LLMs' output logits, which is more costly than computing the similarity score. This could limit the method's practical application, especially when dealing with large datasets. I would expect a runtime comparison between their method and baselines, or to discuss potential optimizations to reduce computational cost.
- **Single Prompt Optimization**: Optimizing adversarial documents based on only a single prompt (“Is this text unintelligible?”) restricts their robustness.
- **Insufficient Evaluation of LLM Detection Evasion**: One of the three “naturalness” filtering prompts (“Is this text unintelligible?”) is identical to the attacker's prompt, and the other two are semantically similar. This resembles a "data leakage" situation, in my opinion. The perplexity-based filtering is also the case (the attacker and defender both use GPT-2). I expect a more comprehensive evaluation using a wider variety of prompts and different LLMs to accurately determine the method's ability to evade detection.
- **Generalizability across Different Retrievers**: Given the relatively low transferability of adversarial decoding across different retrievers, more experiments on different retrievers are needed to verify the effectiveness of the proposed method.

Presentation:

- Figure 1 is too large, in my opinion. It might be better to present two figures (e.g., one for trigger attack and one for non-trigger attack) horizontally.
- The table's caption should be placed before the table.

**Questions:**

- Q1: In line 378, The author stated "Table 2 shows that evasion of detection transfers to two other prompts and another LLM". It is confusing as table 2 does not seem to include the results for other prompts and LLMs. So where is the result for evasion of detection on other prompts and LLMs?

- Q2: In experiment setup, The author said "To evaluate 'naturalness' of adversarial and real documents, we prompt GPT-4o and LLaMA-3.1-8B with these prompts". But where is the result of GPT-4o filtering?

- Q3: In table 2, at the same threshold, increasing the width of the beam search actually increases the true positive rate of LLM-based naturalness filtering (0.5 -> 0.7), which means more adversarial document is filtered. This is very strange to me. In my opinion, increasing the width of beam search should be able to find better solutions (i.e., more stealthy adversarial documents) and therefore less likely to be detected.

---

> ### Author Response · Authors · 2024-11-25
>
> > Dependence on Surrogate LLM
>
> Our method is efficient, taking only about 3 minutes to generate one adversarial document.  Note that all known attacks on retrieval or RAG require only a single or a few adversarial documents for the entire retrieval corpus. Therefore, the attacker never needs to generate many adversarial documents.
>
> We can further apply KV cache to speed up generation. Here is the running time comparison:
>
> | Method | Time (min:sec) |
> |------- | -------|
> | No KV Cache          | 3:46  |
> | KV Cache          | 1:10  |
>
>
> > Single Prompt Optimization
>
> Our evaluation includes experiments showing that generation using a single prompt transfers to other prompts. Moreover, we evaluate generated documents using multiple LLMs (Llama, GPT4o, and Gemma) and multiple prompts.  They evade detection by all of them.
>
> > Insufficient Evaluation of LLM Detection Evasion
>
> Thank you for the suggestions. We have added experiments to use more LLMs for naturalness evaluation and perplexity measurements. Please see “Multiple LLMs for naturalness evaluation” and “Multiple LLMs for perplexity measurement” in General Response.
>
> > Generalizability across Different Retrievers
>
> Thank you for the suggestions.  We have added experiments for two new retrievers.  Please see the “Multiple Retrievers” section in General Response.
>
> > Naturalness Evaluation Questions
>
> Please see the “Multiple LLMs for Naturalness Evaluation” section in General Response.
>
> > Table 2 Beam search Question
>
> Since the current naturalness score of our adversarial texts is sufficiently high to bypass the defense, we cap the naturalness score objective and only further optimize for cosine similarity when we increase the beam size.

---

> > ### Comment · Reviewer_sCy5 · 2024-11-27
> > **comments**
> >
> > Thank you for your response and the additional experiments. I can now see the results of the naturalness evaluation of LLMs with different prompts. However, I still have a few questions:
> >
> > 1. In the "Multiple LLMs for Naturalness Evaluation" section, for each query, how many samples were generated? Were multiple trials conducted?
> > 2. The evaluation prompts still appear semantically similar to me. I would expect to see whether **AdversarialDecoding** can evade detection with prompt less similar to the attack one, such as: *"You are tasked to evaluate the naturalness of the provided text. ... Use the following criteria to assess naturalness: ..."*
> > 3. The authors did not address my question in Q1. While the authors stated that *"Table 2 shows that evasion of detection transfers to two other prompts and another LLM,"* the *Adv* column only displays results for Llama.
> > 4. Regarding the statement: *"We cap the naturalness score objective and only further optimize for cosine similarity when we increase the beam size,"* where is this step reflected in your method? I could not find a corresponding explanation in the paper.

---

### Official Review · Reviewer_NX6r · 2024-11-02

**Soundness:** 2
**Presentation:** 2
**Contribution:** 2
**Rating:** 5
**Confidence:** 3

**Summary:**

This paper investigates the vulnerability of modern retrieval systems that rely on embedding similarity. The authors highlight that existing adversarial methods, such as HotFlip, generate malicious documents with high perplexity, making them easily detectable through perplexity filtering and large language model (LLM) evaluations. To address this, the paper introduces a novel controlled generation technique called AdversarialDecoding, which simultaneously optimizes for embedding similarity and naturalness using a surrogate LLM. This approach produces adversarial documents that maintain low perplexity and appear natural, effectively evading both perplexity-based and LLM-based detection mechanisms. Experimental results on the MS MARCO dataset demonstrate that AdversarialDecoding achieves high poisoning success rates comparable to traditional methods while significantly reducing the likelihood of detection. Additionally, the study explores the limited transferability of these adversarial documents across different encoders, suggesting potential avenues for developing robust defenses. The research underscores the importance of advancing defensive strategies to safeguard retrieval systems against sophisticated adversarial attacks.

**Strengths:**

- The paper introduces a novel controlled generation method named AdversarialDecoding, which uniquely integrates embedding similarity with a "naturalness" constraint. By leveraging a surrogate large language model (LLM) to compute soft scores, the method simultaneously optimizes for semantic relevance and textual naturalness.

- The methodology presented in the paper is robust and meticulously designed. The authors conduct comprehensive experiments using the MS MARCO datasets, and give a lot of ablation studies.

**Weaknesses:**

- The writing in this paper could be **improved a lot**. Firstly, each formula lacks numbering. Additionally, the citation format in lines 251-258 seems off. Moreover, line 123 ends with a comma, and line 130 lacks a period. These issues are quite frequent in the article, which suggests a need for more attention to detail.

- In some experiments, using llama3.1-8b as both the adversarial decoding model and the naturalness evaluation model could raise concerns about fairness. This is because llama3.1-8b might be biased towards the data it generates itself. Besides, could you explain why you use GPT2 to measure the perplexity of generated adversarial documents rather than GPT3 or other LLMs?

- The selected baselines are limited. Hotflip is an early character-level adversarial attack method, but since then, many more effective attack algorithms[1] have been developed, whether at the word, character, or sentence level. These newer methods often result in much higher fluency.

- Adding some additional human evaluations would be valuable.

**References:**

[1] https://github.com/thunlp/TAADpapers

**Questions:**

See the weaknesses.

---

> ### Author Response · Authors · 2024-11-25
>
> > presentation
>
> thanks for the suggestions! In the next revision, we will improve the presentation.
>
> > perplexity and naturalness evaluation
>
> Please see “Multiple LLMs for Perplexity Measurement” and “Multiple LLMs for Naturalness Evaluation” in General Response.
>
> > Baseline comparison
>
> Please see “Multiple LLMs for Naturalness Evaluation” in General Response.
>
> > Human Evaluation
>
> In all existing retrieval and RAG attacks, adversarial documents are added to corpora consisting of thousands or millions of documents.  Human inspection of every document is not a feasible defense in these scenarios, therefore we focus on automated defenses.

---

### Official Review · Reviewer_SPBo · 2024-11-04

**Soundness:** 2
**Presentation:** 3
**Contribution:** 2
**Rating:** 3
**Confidence:** 5

**Summary:**

This work points to an issue in previous retrieval poisoning attacks—detectability via fluency-based defenses—and addresses it by proposing a new method. Specifically, it introduces a black-box attack that uses beam search to generate adversarial passages following both the retrieval objective, and text perplexity and naturalness (i.e., the level of naturalness as judged by an auxiliary LLM) penalties. The attack shows comparable performance with prior work, while it is arguably harder to detect by standard fluency-based defenses.

**Strengths:**

* The work identifies–and clearly states–a limitation in existing retrieval attacks, and proposes a method to address it.

* As the evaluation shows, the proposed attack is harder to detect through the proposed fluency-based detectors (including perplexity and naturalness), while attaining comparable attack success to prior attacks, which further emphasizes the vulnerability of retrieval.

**Weaknesses:**

**Novelty.** The work’s novelty focuses on the naturalness of adversarial documents generated by the new method. However:

* The main novelty of the method—enriching  the objective with LLM logit-based naturalness score (Sec. 5.2)—lacks a convincing presentation (see more details below) and its current evaluation might be biased (more below), especially in light of the repetitive text shown in qualitative samples.
* It was previously shown that the discrete text optimization’s (e.g., in LLM jailbreaks) trade-off between attack success and text fluency [1] can be mitigated (e.g., [2], [3]). Specifically, similarly to this work, Sadasivan et al. [3] show LM-logit-based beam search to produce fluent text attacks. Thus, it is unclear whether this work offers a significant delta w.r.t. to previous work.

**Method.** Since it is introduced as a core contribution, it would be helpful to elaborate on the soft naturalness score component in the method. This, for example, could be done by reporting on a study of alternative scores (if such were considered), or exploring the correlation between the soft naturalness score with naturalness of text.

**Threat Model.** It is unclear why an attacker would aim to generate unconstrained documents (potentially meaningless) and promote their retrieval. For example, In the trigger attack is motivated by the potential of “spreading adversarial content” (line 112), although, to my understanding, such content is not necessarily contained in the generated documents.

**Evaluation.** As the work’s contribution focuses on the “naturalness” of the generated documents, it would be helpful to strengthen the evaluation:

* **Perplexity Filtering (Sec 7.2).** As GPT2 is a relatively dated and weak LLM (line 329), it would be helpful to additionally calculate documents’ perplexity using stronger LLMs (e.g., Gemma2-2B or others), and show that the method is robust to such filtering.
* **Naturalness Filtering (Sec. 7.3).** It seems that the naturalness evaluation for the non-basic attack (“Adv”) is largely done using the same LLM (Llama) used in the attack. A stronger result would be to show the generated documents are robust to naturalness filtering of different, strong, LLMs. Alternatively, one could ask LLMs for a score in a large range (e.g., 1-10), as the current prompt (asking for binary score) could possibly bias the LLM’s answer Another option is reporting on a user study of their naturalness.
* **Evaluated Model(s).** The paper evaluates the attacks against a __single__ retrieval model (namely, Contriever [4]). It should be noted that the evaluated dataset (MS-MARCO) is out-of-training-distribution for this model (Contriver was not trained on MS-MARCO [4], as opposed to most text encoders), and it was previously observed to be exceptionally vulnerable to such retrieval attacks [5]. Thus, it would be helpful to validate the results on additional models.

**Presentation.** Some presentation-related comment and nits:
* Sec. 7.3: It would be helpful to state in the text (besides the table caption) that the evaluated attack is trigger attack.
* Fig. 1: The figure would be easier to interpret if the y-axis ticks would match the (pre-log) values from text.
* Algorithm 1: As LLM_{logits},LLM_{naturalness} and \lambda are all part of the algorithm parametrization, it would be clearer if these were included in the Input.
* Algorithm 1, line 23: Shouldn’t `k` be `m` (in the comment)?

**References:**

[1] Baseline Defenses for Adversarial Attacks Against Aligned Language Models; Jain et al. 2023.

[2] FLRT: Fluent Student-Teacher Redteaming; Thompson & Sklar, 2024.

[3] Fast Adversarial Attacks on Language Models in One GPU Minute; Sadasivan et al., ICML 2024.

[4] Unsupervised Dense Information Retrieval with Contrastive Learning; Izacard et al., TMLR 2022.

[5] Poisoning Retrieval Corpora by Injecting Adversarial Passages; Zhong et al., EMNLP 2023.

**Questions:**

* It seems that the attack algorithm is given a prefix prompt (per line 186), however, unless I missed it, it is not mentioned in the text (Sec. 5). Could you clarify what is the role of this prefix and how it is chosen?
* Results in Sec 7.4, Table 3 (e.g., HotFlip, Top-20, 0.01; with 500 passages), seem to contradict those originally reported by Zhong et al. 2023 [5] on the same dataset and model (e.g., Top-20, 98% with 50 passages). It would be helpful to clarify this discrepancy.
* In line 243, it is mentioned that the no-trigger attack is tested against 1K queries. Are these disjoint from the 50K attacked set (similar to the trigger attack’s evaluation)?

---

> ### Author Response · Authors · 2024-11-25
>
> > Novelty: comparison with previous beam search based works
>
> Our contribution is to show that simply optimizing for low perplexity doesn't produce natural texts. We propose a metric to detect these low-perplexity but unnatural texts, and also propose a method to generate actually natural texts.
> Please see the “New baseline benchmarks” section in General Response for more information.
>
> > Method: soft naturalness score
>
> The purpose of the naturalness score is to guide the optimization of adversarial documents. It might not be well-calibrated but, as our experiments show, it is sufficient to produce adversarial documents that are not detected as unnatural by any currently known method.
>
> > LLM naturalness evaluation
>
> Our apologies if this was not clear in the original submission, but we use *both* Llama and GPT4o for naturalness evaluation. We have now added Gemma – please see “Multiple LLMs for Naturalness Evaluation” section in General Response.
>
> > Perplexity Filtering
>
> Thank you for the suggestion! We have added perplexity measurements with three more LLMs, Llama, Gemma, and Qwen. All experiments show that the perplexity distribution of texts produced by AdversarialDecoding has high overlap with normal texts. Please see the “Multiple LLMs for Perplexity Measurement” section in General Response.
>
> > Threat model
>
> In existing RAG attacks (see related work in our paper), adversarial documents consist of two parts: the sub-document responsible for retrieval (this is what we focus on) and a *separate* sub-document responsible for influencing generation.  Our method for generating retrieval sub-documents can thus be combined with existing attacks on RAG and other systems downstream of retrieval.
> It is easy to change the optimization objective to maximize the attack success rate of an adversary-chosen prefix + our optimized adversarial text. We have added experiments to demonstrate the feasibility of this attack, setting the prefix to “[Trigger] is awful”. Please see the “prefix attack” section in General Response.
>
> > Multiple Retrievers
> Thanks for your suggestions! We have added evaluations on more retrievers. Please see the “Multiple Retrievers” section in General Response.
>
> > prefix
>
> Since we are sampling from an LLM, we need to start from some text. For the trigger attack, we use the prefix “tell us a story about [trigger]” as a hint to search for a sentence related to the [trigger]. We show in the GPT4o baseline that simply generating a sentence with this prefix doesn't work. For the no-trigger attack baseline, we use the prefix “tell us a story” to prepare the LLM to generate diverse texts.
>
> > HotFlip baseline
>
> We set the number of tokens we use to 32 instead of 50 (used in [5]).  HotFlip generates texts with high, extremely abnormal perplexity, which are very easy to detect.  By contrast, our objective is stealthiness.
>
> > Test query
>
> Thanks for pointing out a problem in our evaluation.  We did not split the datasets because these adversarial texts seldomly suffer from overfitting.  We fixed the issue and re-ran the evaluation on 1K disjoint queries from the test query set. Here is the updated Table 3:
>
> | Method                     | Top-1 | Top-5 | Top-10 | Top-20 | Top-100 |
> |----------------------------|-------|-------|--------|--------|---------|
> | BeamSearchHotflip          | 0.00  | 0.01  | 0.01   | 0.02   | 0.05    |
> | PerplexityLLMOptimizer     | 0.08  | 0.20  | 0.27   | 0.37   | 0.59    |
> | NaturalnessLLMOptimizer    | 0.02  | 0.06  | 0.09   | 0.13   | 0.31    |
>
> > Presentation
>
> Thanks for the suggestions!  In the next revision, we will improve the presentation.

---

> > ### Comment · Reviewer_SPBo · 2024-11-27
> > **Thank you & follow-up comments**
> >
> > Thanks for addressing some of the raised issues; including evaluating perplexity against different models (which does present stronger results), adding the prefix attack and making clarifications. However, there are major issues remain unresolved:
> >
> > 1. **Comparison with Fluency-Optimized LLM Attacks.** The proposed comparison with “beam-search work” is a step forward toward highlighting the motivation for involving naturalness in attacks. However, the evaluation should be made more precise and fair: currently it compares apples (attacks against LLMs) to oranges (attacks against retrievers). Additionally, it is unclear how similar are the settings of the three attacks (e.g., a different length adversarial suffix is expected to affect the naturalness, as well as other parameters), and it is possible that LLMs refuse to answer some prompts (due to potential harm, as these are originally jailbreak prompts). Finally, presenting attack examples for each evaluated attack (e.g., to showcase the lack of naturalness of existing ones), can further strengthen the paper.
> > 2. **Evaluation of Naturalness.** The naturalness metric (Section 6) still lacks a convincing justification. Specifically, I am concerned binary questions to LLMs might bias evaluation. To rule this out, I recommend further validating the findings through querying LLMs with non-binary scores (e.g., 1-10), in addition to a possible online study with human subjects.
> > 3. **Evaluation on multiple retrievers.** Validating results on multiple retrievers is a valuable step. However, as only a narrowly scoped experiment is run on the newly added retrievers, it is unknown which of the findings generalize beyond the Contriever model (which, as noted in the original review, is trained on out-of-distribution data).
> > 4. **Threat Model.** As mentioned in the original review, it seems that most of the evaluation (besides the new prefix attack) does not adhere to the threat model’s assumptions. Accordingly, either the threat model needs to be adjusted, or all experiments should be repeated with the prefix attack.
> > 5. **HotFlip Baseline.** A clarification of the results in Table 3 would be helpful. The response states that the ASR drop in the HotFlip evaluation is due to optimizing fewer tokens than Zhong et al. 2023, albeit using x100 poisoning rate. However, Zhong et al.’s results demonstrate 98% ASR, as opposed to the 1% ASR in Table 3. Is it possible that a certain defense is applied but not explicitly mentioned? If not, it would be helpful to double-check and discuss what causes unconstrained HotFlip to provide _worse_ results than the proposed fluency-constrained attack.
> >
> > Last, I’d like to note that I found the response somewhat hard to follow. Particularly, upon reading the response, it is not immediately clear which changes have been already made in the PDF and which will only be made in future revision (and if so where). A clarification about the updates (both past and planned changes) and a diff would be extremely helpful.

---

### Official Review · Reviewer_wZCy · 2024-11-04

**Soundness:** 2
**Presentation:** 3
**Contribution:** 3
**Rating:** 5
**Confidence:** 4

**Summary:**

This paper focuses on the problem of retrieval poisoning in modern retrieval systems. Firstly, it points out the limitations of existing methods (such as HotFlip and COLD) in generating adversarial documents. The documents generated by HotFlip have a relatively high perplexity and are easily detected; while COLD fails to generate useful texts under adversarial constraints. Then, this paper proposes a new controlled generation technique, which combines an adversarial objective (embedding similarity) with a "naturalness" objective calculated based on an open-source surrogate language model (LLM). The generated adversarial documents are difficult to be detected by perplexity filtering or other LLMs without generating a large number of false positives. This method has been evaluated in different scenarios such as trigger attacks and no-trigger attacks, using the MS MARCO dataset. In terms of poisoning efficacy and the naturalness of generated documents, it is superior to previous methods, but still has some limitations, such as poor transferability across encoders and the need for more research on defenses.

**Strengths:**

1. The proposed adversarial decoding method is a novel controlled generation technique that comprehensively considers embedding similarity and naturalness and effectively addresses the deficiencies of existing methods.
2. Experiments were conducted using the large-scale MS MARCO dataset, comparing different generation methods and considering two scenarios: trigger attacks and no-trigger attacks.

**Weaknesses:**

1. The transferability of adversarial documents between different encoders is poor, which limits the universality of the method.
2. It depends on LLM and does not consider the situation of LLM hallucination. In addition, the use of LLM needs to consider the efficiency and effectiveness of the attack.
3. The experiments are not sufficient. The experiment only considers one retriever, contriver, and other retrievers need to be compared. At the same time, the baselines need to be increased (for example, PoisonedRAG in the references).

**Questions:**

1. How to further enhance the attack effect while improving the naturalness of adversarial documents?
2. For different types of retrieval systems and application scenarios, does this method need to be specifically adjusted?
3. How to better understand and quantify the "naturalness" indicator in order to more accurately evaluate the generated adversarial documents? Is it reasonable to rely solely on perplexity?
4. How to consider the hallucination and efficiency problems caused by the auxiliary LLM?

---

> ### Author Response · Authors · 2024-11-25
>
> # Weaknesses:
> > transferability
>
> As we show in our paper, the baselines also transfer poorly.  Therefore, this limitation is not specific to our method but a generic issue with attacks on encoders.  One possible solution is to integrate multiple encoders into the optimization objective.
>
> > It depends on LLM and does not consider the situation of LLM hallucination. In addition, the use of LLM needs to consider the efficiency and effectiveness of the attack.
>
> To avoid LLM hallucination, we use multiple LLMs in the evaluation stage (Llama and GPT4o in the original submission; we have now added Gemma 2). Please refer to "Multiple LLMs for naturalness evaluation" section in Genereal Responses.
>
> Our method is efficient, taking about 3 minutes to generate one adversarial document.  Note that all non-trivial attacks on retrieval or RAG require only a single or a few adversarial documents for the entire retrieval corpus. Therefore, the attacker never needs to generate many adversarial documents. We also discuss the efficiency of our method below.
>
> > PoisonedRAG comparison and multiple retrievers
>
> Thank you for the suggestions!
>
> PoisonedRAG focuses on poisoning retrieval results for a specific query.  We demonstrate how to generate natural-looking adversarial documents for a wide range of queries (same setting as Zhong et al., EMNLP 2023).
>
> We have added experiments with two additional retrievers. Please see the “Multiple Retrievers” section in General Response.
>
> # Questions:
> > How to further enhance the attack effect while improving the naturalness of adversarial documents?
>
> As our experiments show, baseline attack success rate is already high. Enforcing the naturalness constraint inevitably restricts the search space, which the maximum achievable ASR. For naturalness, we show that our documents evade all currently known defenses.
>
> > For different types of retrieval systems and application scenarios, does this method need to be specifically adjusted?
>
> We can incorporate several different retrieval objectives.  Please see our new experiments for other retrievers.
>
> > How to better understand and quantify the "naturalness" indicator in order to more accurately evaluate the generated adversarial documents? Is it reasonable to rely solely on perplexity?
>
> We show in our paper that perplexity is not enough to define naturalness.  In fact, one of our contributions is an effective method to detect adversarial documents that have low perplexity.
>
> The naturalness score is our proposed optimization objective *in addition to perplexity*.  We also propose a systematic way to evaluate naturalness by using multiple LLMs and different prompts.
>
> > How to consider the hallucination and efficiency problems caused by the auxiliary LLM?
>
> Please see response to the second weakness point above.

---

### Official Review · Reviewer_QPvX · 2024-11-04

**Soundness:** 2
**Presentation:** 1
**Contribution:** 1
**Rating:** 3
**Confidence:** 4

**Summary:**

This paper introduces a beam-search-based adversarial attack method for RAG, designed to produce fluent text with sentence embeddings closely matching a target embedding. Experimental results demonstrate that this approach effectively bypasses perplexity-based filtering and achieves a comparable attack success rate to HotFlip baselines.

**Strengths:**

The proposed method is straightforward and easy to follow.

Experiments are conducted on a recent dataset and compared against current baselines.

**Weaknesses:**

(1) The proposed attack method relies on simple prompts like “Tell me a story about” to generate documents from scratch. This approach raises concerns about practical applicability, as real-world malicious users typically aim to inject misinformation. It is unclear how the proposed method would effectively introduce misinformation in a RAG setting.

(2) The paper lacks experiments demonstrating how the proposed method impacts the end-to-end performance of RAG systems, such as downstream QA performance.

(3) The novelty of this work is limited, as similar approaches have been applied in slightly different contexts. For example, beam search algorithms have been widely used in adversarial attacks [1][2]. The paper should discuss these related works and emphasize its unique contributions beyond altering the beam search optimization objective.

> [1] Zhao et. al., Generating Natural Language Adversarial Examples through An Improved Beam Search Algorithm

> [2] Liu et. al., A More Context-Aware Approach for Textual Adversarial Attacks Using Probability Difference-Guided Beam Search

(4) The paper claims black-box access to the embedding encoder. However, given the assumption that embeddings can be accessed repeatedly, one could calculate gradients numerically, making the black-box claim somewhat overstated.

(5) Some other minor issues:
- Please use `\citep` and `\citet` in Latex properly.
- The paper uses the gendered pronoun "his" for attackers (Line 109), which could be avoided.
- The paper contains several grammatical mistakes
- Notation definitions lack precision and could be simplified. For example, `P_n`​ is defined as a retrieval corpus but actually represents benign documents. The subscript `n` could be omitted.

**Questions:**

(6) Why not use a weighted sum of embedding similarity and perplexity instead of introducing an extra model?

(7) Why are only true positives and false positives considered for defense? Would false negatives not be equally important?

(8) Is the LLM naturalness evaluator used during the attack aligned with the one used in the evaluation?

---

> ### Author Response · Authors · 2024-11-25
>
> > prefix and misinformation
>
> Since we are sampling from an LLM, we need to start from some text. For the trigger attack, we use the prefix “tell us a story about [trigger]” as a hint to search for a sentence related to the [trigger]. We show in the GPT4o baseline that simply generating a sentence with this prefix doesn't work.
>
> For the no-trigger attack baseline, we use the prefix “tell us a story” to prepare the LLM to generate diverse texts. For the results, please see the “prefix attack” section in General Response.
>
> We have also added the attack setup for which we spread information by inserting misinformation before our text. Please refer to "Prefix attack" section in General Response.
>
> > end-to-end RAG performance
>
> We work in the same setting as Zhong et al. “Poisoning Retrieval Corpora by Injecting Adversarial Passages” (EMNLP 2023), focusing on retrieval.
>
> In existing RAG attacks (see related work in our paper), adversarial retrieval is separate from adversarial generation.  Adversarial documents consist of two parts: the sub-document responsible for retrieval (this is what we focus on) and a separate sub-document responsible for influencing downstream generation.  Our method for generating retrieval sub-documents can thus be combined with existing attacks on RAG and other systems downstream of retrieval.
>
> > novelty and weighted sum
>
> These papers apply beam search to sentence editing, not generation.
>
> As we show in our BasicAdversarialDecoding setting, beam search alone does not produce natural text. Existing beam search methods produce low-perplexity adversarial documents that are detectable – using the method proposed in this paper – as unnatural.
>
> Our contributions are new methods for (a) detecting adversarial texts that have low perplexity, yet are unnatural (this includes documents produced by previous beam search methods), and (b) incorporating the naturalness objective into beam search.
>
> We have also added experiments to compare with two recent beam search-based adversarial generation methods.  Please see the “New baseline benchmarks” section in General Response.
>
> > black-box
>
> We use the standard definition of black-box access from the security literature: the attacker can query the model freely to get the outputs, but does not have access to the weights of the model.
>
> > llm naturalness evaluator robustness
>
> Please refer to “Multiple LLMs for Naturalness Evaluation” section in General Response.
>
> > false negative
>
> We provided true positives.  By definition, false negatives = 1 - true positives.
>
> > presentation
>
> Thanks for your suggestions, we will make the changes in next version.

---

> > ### Comment · Reviewer_QPvX · 2024-12-02
> > **Follow up comments**
> >
> > Thank you for your response. I appreciate the effort in addressing my concerns; however, I remain unconvinced about the practical relevance of the attack settings. While I understand that the work builds on Zhong et al. (EMNLP 2023, Short Paper), I struggle to envision a realistic scenario where this type of attack could pose significant real-world challenges.
> >
> > There are several issues with the proposed setting. For instance, why would an attacker have control over the sub-documents used in RAG? Furthermore, why is the alignment between the sub-document and the main document for downstream tasks not validated, especially when fluency checks are performed?
> >
> > I encourage the authors to critically evaluate the assumptions and settings adopted from previous work.
> >
> > I will maintain my original score.

---

### Author Response · Authors · 2024-11-25
**General Response**

# New baseline benchmarks

The state of the art for producing fluent adversarial texts are BEAST and FLRT.  They produce text that has low perplexity but is detected as unnatural by the detection method proposed in this paper.

We applied the naturalness evaluation metric proposed in this paper to all 10 examples provided  by BEAST [1] and all 41 examples provided by FLRT [2].  We then asked GPT4o, Llama-3.1 8B, and Gemma 2 9B these three questions:

    "Is this text meaningless?"

    "Is this text gibberish?"

    "Is this text unintelligible?"

The score for a given document is the total of all LLM-prompt combinations where the answer is “No”, thus scores range from 0 to 9.  Here are the scores for BEAST and FLRT texts:

BEAST: `[5, 0, 0, 1, 0, 2, 0, 0, 0, 0]`

FLRT: `[0, 0, 0, 0, 0, 3, 0, 0, 0, 1, 1, 0, 0, 3, 5, 0, 0, 0, 0, 4, 0, 3, 5, 0, 0, 0, 0, 0, 0, 0, 0, 0, 0, 0, 0, 0, 0, 0, 0, 0, 0]`

This clearly shows that BEAST and FLRT documents are recognized as unnatural and fail to evade detection.  By contrast, here are the scores for texts produced by our method:

AdvDecoding: `[9, 6, 9, 6, 6, 3, 6, 9, 6, 9]`

# Multiple Retrievers

We have added experiments with two additional retrievers.

| Model                                    | Top-1 | Top-3 | Top-5 | Top-10 | Top-100 |
|------------------------------------------|-------|-------|-------|--------|---------|
| sentence-transformers/sentence-t5-base  | 0.08  | 0.12  | 0.15  | 0.19   | 0.35    |
| sentence-transformers/gtr-t5-base       | 1.00  | 1.00  | 1.00  | 1.00   | 1.00    |
| facebook/contriever                     | 0.04  | 0.23  | 0.29  | 0.43   | 0.83    |


While success rate varies from retriever to retriever, documents generated by our method achieve non-trivial success rates against all of them that enable practical attacks.



# Multiple LLMs for perplexity measurement

We have added the perplexity measurement by Llama, Gemma and Qwen. Please see the updated PDF. We have updated section 7.2 and Figure 1 to reflect the changes. We show that our methods evade perplexity detection by all LLMs.


# Multiple LLMs for naturalness evaluation

In our naturalness evaluation, we use both Llama-3.1 8B (the LLM we use for generation) and GPT4o (which we do *not* use for generation). We also added evaluation experiments with Gemma 2 9B. We use the same method as in the “new baseline benchmarks” to compute the naturalness score.

| Query            | GPT2BasicAdversarialDecoding | LlamaBasicAdversarialDecoding | AdversarialDecoding |
|-------------------|-----------------------------|-------------------------------|---------------------|
| spotify           | 0                           | 0                             | 9                   |
| xbox              | 0                           | 0                             | 6                   |
| lebron james      | 0                           | 0                             | 9                   |
| amazon            | 0                           | 0                             | 6                   |
| iphone            | 0                           | 0                             | 6                   |
| netflix           | 0                           | 0                             | 3                   |
| BMW               | 0                           | 0                             | 6                   |
| Marilyn Monroe    | 0                           | 0                             | 9                   |
| nfl   | 0                           | 3                             | 6                   |
| Olympics    | 0                           | 0                             | 9                   |


This shows that documents generated by our method achieve high naturalness scores with all three LLM evaluators.



# Prefix attack
It is easy to change the optimization objective to maximize the attack success rate of an adversary-chosen prefix + our optimized adversarial text to spread misinformation. We have added experiments to demonstrate the feasibility of this attack, setting the prefix to “[Trigger] is awful”.
|                       | Top-1 | Top-3 | Top-5 | Top-10 | Top-100 |
|-----------------------|-------|-------|-------|--------|---------|
| Adversarial Decoding  | 0.21  | 0.29  | 0.36  | 0.46   | 0.75    |

References:

[1] Fast Adversarial Attacks on Language Models in One GPU Minute; Sadasivan et al., ICML 2024.

[2] FLRT: Fluent Student-Teacher Redteaming; Thompson & Sklar, 2024.

---

### Meta-Review · Area_Chair_wceg · 2024-12-20

**Metareview:**

This work identifies a challenge in earlier retrieval poisoning attacks—their vulnerability to detection by fluency-based defenses—and proposes a novel solution. It presents a black-box attack that employs beam search to generate adversarial passages, optimizing for both the retrieval objective and penalizing text perplexity and unnaturalness (as assessed by an auxiliary LLM). The proposed attack achieves performance on par with prior methods while being significantly more resistant to detection by standard fluency-based defenses.

Strength:
- identification of previous methods limitation and proposed a method inspired by this vulnerability.

Weaknesses (still remain after the rebuttal):
  - There are several issues with the proposed setting. For instance, why would an attacker have control over the sub-documents used in RAG? Furthermore, why is the alignment between the sub-document and the main document for downstream tasks not validated, especially when fluency checks are performed?
The evaluation lacks fairness, comparing methods for LLMs and retrievers with differing settings, such as adversarial suffix lengths. The absence of attack examples limits the clarity of naturalness evaluation differences.

- The reliance on binary prompts for evaluation may introduce bias. Non-binary scoring or human evaluation would strengthen the validity of the naturalness claims.

- The findings are primarily tested on the Contriever model, which is trained on out-of-distribution data, making generalization uncertain.

- The evaluation does not consistently adhere to the stated threat model, particularly for non-prefix attacks. Adjustments or expanded experiments with the prefix attack are needed.

- The HotFlip results in Table 3 significantly deviate from previously reported benchmarks, lacking an explanation for this inconsistency.

**Additional Comments On Reviewer Discussion:**

The reviewers of this submission are extremely responsive, but many limitations and weaknesses remain after the rebuttal. I suggest that the authors address these issues before the next submission.

---

### Decision · Program_Chairs · 2025-01-22

Reject